# Overexpression of *DGAT2* Regulates the Differentiation of Bovine Preadipocytes

**DOI:** 10.3390/ani13071195

**Published:** 2023-03-29

**Authors:** Pan-Pan Guo, Xin Jin, Jun-Fang Zhang, Qiang Li, Chang-Guo Yan, Xiang-Zi Li

**Affiliations:** 1Guangdong Provincial Key Laboratory of Large Animal Models for Biomedicine, School of Biotechnology and Health Sciences, Wuyi University, Jiangmen 529020, China; 2Healthcare International Innovation Institute, Jiangmen 529020, China; 3Engineering Research Centre of North-East Cold Region Beef Cattle Science & Technology Innovation, Ministry of Education, Department of Animal Science, College of Agriculture, Yanbian University, Yanji 133002, China; 4Laboratory Animal Center, Yanbian University, Yanji 133002, China

**Keywords:** adipocytes, *DGAT2*, overexpression, transcriptome, triacylglycerol

## Abstract

**Simple Summary:**

Beef with rich intramuscular fat deposits is considered to be of high quality, and triacylglycerols are a major component of intramuscular fat. The synthesis of triacylglycerols is regulated by diacylglycerol O-acyltransferase 2 (*DGAT2*). This study focused on the regulatory mechanism of *DGAT2* in the differentiation of preadipocytes from Yanbian cattle and its role in lipid metabolism-related signalling pathways. Bovine preadipocytes were infected by constructing an overexpression adenovirus vector and an interfering adenovirus vector. Differentially expressed genes in bovine preadipocytes were analysed using RNA sequencing and genome databases. The results showed that during the differentiation of bovine preadipocytes, high expression of *DGAT2* promoted lipid droplet formation, triglyceride content, and expression of adipogenesis genes at the messenger RNA level. Transcriptome analysis identified 208 differentially expressed genes between DGAT2-overexpressing preadipocytes and normal cells. These results indicate that *DGAT2* plays an important role in the regulation of bovine fat metabolism and provides a theoretical basis for the production of high-quality marbled beef.

**Abstract:**

Triacylglycerols (TAGs) are a major component of intramuscular fat. Diacylglycerol O-acyltransferase 2(*DGAT2*) expression determines the rate of TAG synthesis. The purpose of this study was to investigate the role of *DGAT2* in the differentiation of Yanbian cattle preadipocytes and lipid metabolism-related signalling pathways. Bovine preadipocytes were infected with overexpression and interfering adenovirus vectors of *DGAT2*. The effects on the differentiation of Yanbian cattle preadipocytes were examined using molecular and transcriptomic techniques, including differentially expressed genes (DEGs) and Kyoto Encyclopaedia of Genes and Genomes (KEGG) pathway analysis. *DGAT2* overexpression significantly increased (*p* < 0.05) intracellular TAG, adiponectin, and lipid droplet (LD) contents. Moreover, it upregulated (*p* < 0.05) peroxisome proliferator-activated receptor γ *(PPARγ)*, CCAAT/enhancer binding protein *α*, and fatty acid binding protein 4 mRNA expression. In contrast, *DGAT2* knockdown reduced intracellular TAG and LD content and downregulated (*p* < 0.05) *C/EBPβ*, mannosyl (alpha-1,3-)-glycoproteinbeta-1,2-N-acetylglucosaminyltransferase, lipin 1,1-acylglycerol-3-phosphate O-acyltransferase 4, and acetyl-CoA carboxylase alpha mRNA expression. Between DGAT2-overexpressing preadipocytes and normal cells, 208 DEGs were identified, including 106 upregulated and 102 downregulated genes. KEGG pathway analysis revealed DEGs mainly enriched in PPAR signalling and AMP-activated protein kinase pathways, cholesterol metabolism, and fatty acid biosynthesis. These results demonstrated that *DGAT2* regulated preadipocyte differentiation and LD and TAG accumulation by mediating the expression of adipose differentiation-, lipid metabolism-, and fatty acid synthesis-related genes.

## 1. Introduction

The beef cattle industry plays an important role in China’s animal husbandry sector and is a source of income. The production of high-quality beef with rich intramuscular fat deposits, such as “Snowflake beef,” could improve the profitability and prospects of cattle fattening operations. The aim of any fattening operation is to produce meat with superior quality characteristics, including tenderness, shear force, and marbling [1]. These quality traits are directly or indirectly related to the intramuscular fat content of beef [2], indicating the importance of intramuscular fat content in determining meat quality.

Intramuscular fat is mainly located in skeletal muscle fibres and stored in lipid droplets (LD). The composition of LD is mainly triglycerides (TAGs) and cholesterol esters [3]. TAGs are the most important form of energy storage in mammals and are synthesised in most cells through the glycerophosphate pathway (Kennedy pathway) and in specialised cells through the glycerol monoacyl pathway [4]. These two pathways synthesise TAG using diacylglycerol and fatty acid acyl groups in a reaction catalysed by diacylglycerol O-acyltransferase (*DGAT*) [5]. Studies on livestock have shown that *DGAT* is widely expressed in mammal tissues and is closely related to livestock reproductive characteristics and other key economic traits, such as carcass and meat quality [6]. *DGAT* has two subtypes (*DGAT1* and *DGAT2*) with significantly different membrane topologies [7] that play different roles in mammalian TAG metabolism. *DGAT1* knockdown reduced TAG synthesis in mice [8], whereas *DGAT2* knockdown resulted in the death of mice after birth, with a >90% reduction in TAG concentration [9]. These findings indicate that *DGAT1* plays a major regulatory role in energy metabolism, whereas *DGAT2* plays an important role in TAG synthesis and storage.

However, the molecular mechanism through which *DGAT2* regulates preadipocyte differentiation and lipid metabolism in cattle is poorly understood. We hypothesised that *DGAT2* induces preadipocyte differentiation. Here, we infected bovine preadipocytes by constructing an overexpression adenovirus vector (Ad–DGAT2) and an interfering adenovirus vector (sh–DGAT2), and molecular and transcriptomic techniques were used to explore the mechanism of action of *DGAT2* in lipid metabolism and preadipocyte differentiation in Yanbian cattle.

## 2. Materials and Methods

### 2.1. Bovine Preadipocytes Isolation, Culture, and Differentiation

Preadipocytes were isolated from subcutaneous adipose tissue obtained from the back of 18-month-old Yanbian cattle using the collagenase digestion method [10]. Briefly, the adipose tissue fragments of Yanbian cattle were sterilised with 75% alcohol and stored in a 15-mL centrifuge tube containing 1% phosphate-buffered saline (PBS; Gibco, Thermo Fisher Scientific, Waltham, MA, USA). Thereafter, the cells (5 × 10^4^) were inoculated in DMEM (Gibco, Thermo Fisher Scientific, Waltham, MA, USA) containing 10% foetal bovine serum (FBS; Gibco, Thermo Fisher Scientific, Waltham, MA, USA) and cultured at 37 °C under a 5% CO_2_ atmosphere. The culture medium was changed every 48 h. At 80% confluency, the cells were infected with DGAT2 overexpression adenovirus (Ad–DGAT2 and Ad–NC) or DGAT2 interference adenovirus (sh–DGAT2 and sh–NC) for 24 h. Preadipocyte differentiation was induced in a differentiation medium (DMEM supplemented with 5% FBS and 100 µM palmitoleic acid). After 96 h of culture, the transfected cells and culture medium were collected for subsequent analysis.

### 2.2. Vector Construction and Adenovirus Packaging

To determine the effect of *DGAT2* in preadipocyte differentiation, we constructed adenovirus vectors (ADVs) overexpressing *DGAT2* and vectors containing short hairpin RNAs (shRNAs) targeting the coding sequence (CDS) region of *DGAT2*. ADV packaging was completed by GenePharma Biotech, Shanghai, China. The primers of the overexpression sequence were designed based on the bovine *DGAT2* gene sequence (GenBank; Accession No. NM_205793.2) and directionally cloned into the *Eco*RI and *Bam*HI sites of the shuttle vector ADV1. Thereafter, the recombinant adenovirus shuttle plasmid (pAdTrack–ADV1–DGAT2) and the backbone plasmid (pGP–Ad–Pac) were prepared using the new adenovirus skeleton gene (missing left ITR, packaging signal, and E1 sequence). After high-purity endotoxin-free extraction, 293A cells were cotransfected with RNAi-Mate to prepare the adenovirus pAd-DGAT2. The negative control was recombinant adenovirus with a green fluorescent protein (Ad–GFP). The virus was purified by a combination of CsCl density-gradient centrifugation-dialysis. The virus titre was checked using the limited dilution method. The adenovirus titres of Ad–DGAT2 were 1 × 10^9^ PFU/mL, and that of Ad–NC was 1 × 10^10^ PFU /mL.

Three shRNAs targeting the CDS region of *DGAT2* (DGAT2–sh–108, DGAT2–sh–320, and DGAT2–sh–687) were designed based on the full-length CDS region of Yanbian cattle *DGAT2* (as shown in Appendix A). Additionally, shRNA–NC that did not target any gene was designed as the control. The sequence information is shown in Table 1. The interfering sequence was cloned into the corresponding site of the shuttle vector ADV1/Eco RI/Bam HI to generate the recombinant adenovirus vector ADV1/U6/CMV-GFP shRNA, and cotransfected with adenovirus backbone plasmid (PGP-Ad-PAC plasmid) into 293A cells. The adenovirus titres of sh–DGAT2 and sh–NC were 1 × 10^9^ PFU/mL and 3 × 10^9^ PFU/mL, respectively.

### 2.3. Optimisation of Conditions for DGAT2 Overexpression/Knockdown

Preadipocytes were inoculated into 96-well plates containing 100 μL of the medium at a density of 3 × 10^3^ cells/well for 24 h and infected with the viruses or 5 μg/mL polybrene at 60~80% cell confluence. The viruses and 5 μg/mL polybrene were added at different multiplicity of infection (MOI) values (stock solution; 10×, 100×, 1000×, and 10,000× dilution rates) to determine the optimal MOI to induce the constructs. Before infection, each well was seeded with 90 μL of fresh culture medium and 10 μL of the virus at different concentration gradients in triplicate for each group. After culturing for 24 h, the wells were replaced with fresh medium, and cell morphology and fluorescence expression were observed under a fluorescence microscope (BX53; Olympus, Tokyo, Japan) to confirm infection. The expression level of the DGAT2 gene was detected by Real-Time PCR to verify the infection efficiency.

### 2.4. RNA Extraction and Quantitative Real-Time PCR (qRT-PCR) Detection

Total RNA was extracted from bovine preadipocytes using TRIzol™ reagent (Thermo Fisher Scientific, Waltham, MA, USA), and RNA integrity was resolved by NanonodropnD-100 spectrophotometer (2000C, Thermo Fisher Scientific, Waltham, MA, USA) and 1% agarose gel electrophoresis. The complementary DNA (cDNA) template was synthesised using the Fastking gDNA Dispelling RT SuperMix kit (Tiangen Biotech, Beijing, China). The qRT-PCR was performed on an Agilent Mx3000/5p Real-Time PCR Detection System (Agilent Technologies, Santa Clara, CA, USA), using 20 μL of SYBR Green SuperReal PreMix Plus (Tiangen Biotech, Beijing, China) containing 10 μL of SYBR, 0.6 μL of up- and downstream primers (10 μmol/L), 1 μL of cDNA template, 7.5 μL of Rnase-free ddH_2_O, and 0.3 μL of Rox. The PCR reaction procedure was predenaturation at 95 °C for 15 min. There were 35 cycles of denaturation at 95 °C for 10 s, annealing at 55 °C for 30 s, and extension at 72 °C for 32 s. Dissolution stage was 95 °C for 15 s and 65 °C for 5 s. The relative expression of the target gene was normalised to that of glyceraldehyde-3-phosphate dehydrogenase (GAPDH; internal control) and calculated using the 2^−△△Ct^ method [11]. Primers for adipose differentiation, lipid metabolism, and fatty acid synthesis-related genes were synthesised by Shenggong Bioengineering Co., Ltd. (Shanghai, China), and their sequences are shown in Table 2.

### 2.5. Western Blotting

Total protein was extracted from transfected cells using a lysis buffer containing 1% phenylmethyl sulfonyl fluoride, and the concentration of total protein was determined using a protein concentration assay Kit (Pierce, Thermo Fisher Scientific Inc.). Protein samples were separated using 10% SDS–PAGE gel at a rate of 20 µg per well and then transferred to polyvinylidene difluoride membranes. The membranes were washed with 1 × TBST and incubated with the following primary antibodies: anti–*DGAT1* (ab100982; Abcam, Cambridge, UK), anti–*DGAT2* (ab237613; Abcam), anti–peroxisome proliferator-activated receptor γ (*PPARγ)* (bs-4509R; Bioss, Beijing, China), anti–CCAAT/enhancer-binding protein α (*C/EBPα)* (bs-1630R; Bioss), anti–*C/EBPβ* (bs-1396R; Bioss), and anti-lipin 1(*LPIN1)* (bs-7533R; Bioss) overnight at 4 °C under constant shaking. This was followed by incubation with the secondary antibody: horseradish peroxidase-conjugated affinipure goat anti-mouse IgG (Jackson, Anhui, China) for 2 h at room temperature (approximately 25 °C) after washing with PBS. The endogenous control was *β-Actin* (AB8226; Abcam). Protein bands were photographed using a multifunctional imaging system (Azure 600; Azure Biosystems, Dublin, CA, USA).

### 2.6. Oil Red O Staining and Triglyceride Determination

The oil red O staining kit (G1262; Solarbio, Beijing, China) was used for oil red O staining. Briefly, the cells were washed with PBS buffer solution, fixed with cytochrome fixative for 30 min, washed with 60% isopropanol, stained with Oil Red O staining solution for 10–15 min, and then counterstained with Mayer haematoxylin solution for 1–2 min. Finally, the cell images were collected under an inverted microscope (IX-73; Olympus) to observe lipid droplets.

The triglyceride content of the cells was determined using the Prelude Triglyceride Determination Kit (Applygen Technologies, Beijing, China). The cells were washed with PBS three times and lysed in lysis buffer (R0010; Solarbio). TAG content was determined by enzymatic colourimetry at 570 nm using a microplate reader (iMark; BIO-RAD, Hercules, CA, USA).

### 2.7. Determination of Adiponectin (ADP) Concentration

After 96 h of induction and differentiation, the cell culture medium was collected and analysed using bovine adiponectin ELISA Kit (Mlbio, Shanghai, China). A microplate reader (iMark, Bio-Rad, HArklues, CA, USA) was used to measure optical density values at 450 nm and generate a standard curve to determine adiponectin concentration.

### 2.8. RNA Sequencing (RNA-seq)

Bovine preadipocytes were infected with Ad–DGAT2, Ad–NC, sh–DGAT2, and sh–NC, with three replicates in each group. Cell samples were collected 96 h after differentiation, and RNA was extracted using 1 mL of Trizol. Shanghai Parsonol Biotechnology Co., Ltd. (Shanghai, China) was commissioned to perform RNA purification, cDNA library construction, and sequencing. The standard of library preparation was sampled with a RIN value of >7.

The samples were sequenced by Illumina^®^ HiSeq TM 2000 (the ultra-long read length of 2 × 150 bp has a better effect of sequence splicing), and the image file was obtained and converted using the sequencing platform software and the genebuild by NCBI. The FASTQ raw data were generated, and each sample was counted separately, including sample name, Q30, percentage of fuzzy bases, and Q20(%) and Q30(%). The sequencing data contained some low-quality reads with joints, which could cause great interference in the subsequent information analysis; hence, the sequencing data was further filtered. We used the HISAT2 (http://ccb.jhu.edu/software/hisat2/index.shtml (accessed on 25 December 2021)) software, and the filtered reads were compared to the reference genome. We evaluated the sequencing quality of the sequencing data volume using a homogeneous degree of gene coverage and saturation analysis. Under ideal conditions, read distribution should be uniform across all expressed genes. We used RSeQC analysis of expression saturation to assess whether the measured amounts of data were sufficient to calculate gene expression levels correctly. We used HTSeq to calculate the read count value of each gene as the original gene expression. The expression quantity was standardised by FPKM. After reading the gene structure annotation information (GTF file), the comparison results were compared with the gene structure and counted according to the Union scheme results. In the reference transcriptome, genes with FPKM >1 are generally considered to be expressed. We used DESeq for differential analysis of gene expression, counted the significant differential gene sets in the results of differential expression analysis, made bar charts for differential genes between different comparison groups, and counted the number of upregulated differential genes and downregulated differential genes in each comparison group. Furthermore, the R language ggplots2 software package was used to draw the volcano map of differentially expressed genes.

### 2.9. Gene Ontology (GO) and Kyoto Encyclopaedia of Genes and Genomes (KEGG) Enrichment Analysis

Differential expression analysis was performed using the deseq software, and genes with log2 foldchange > 1 at *p* < 0.05 were considered as differentially expressed genes (DEGs). Thereafter, the DEGs were mapped to the GO database (http://geneontology.org/ (accessed on 25 December 2021)) using the TopGo software for functional annotations. Additionally, KEGG pathway analysis (http://www.kegg.jp/ (accessed on 25 December 2021)) was performed to identify significantly enriched pathways by the DEGs using the clusterProfiler software. Pathways were considered significantly enriched at *p* < 0.05.

### 2.10. Statistical Analyses

Data analysis and generation of graphs were performed using GraphPad Prism 6.07 (GraphPad Software, La Jolla, CA, USA) and the SPSS Statistical software v19.0 (IBM, Armonk, New York, NY, USA). The 2^−ΔΔCt^ method [11] was used to analyse the qRT-PCR data, and differences among treatments were determined using the Student’s *t*-test. One-way ANOVA was used for statistical comparisons and for determining the difference in TAG and ADP content among treatments. The results were presented as the mean ± standard error of the mean (SEM) from experiments performed in triplicate. The difference was statistically significant at *p* < 0.05, which was used as the criterion to judge the significance of the difference.

## 3. Results

### 3.1. Optimal MOI Value and Infection Efficiency of the Shuttle Viruses

Treatment with 5 μg/mL of polybrene and the adenoviruses at MOI of 100 did not alter the morphology and structure of the preadipocytes (Figure 1A and Figure 2A). Real-time PCR showed that Ad–DGAT2 treatment increased *DGAT2* expression by more than 500 times compared with that in the Ad–NC group (Figure 1B). Compared with the sh–NC control group, DGAT2–shRNA–320 treatment significantly decreased *DGAT2* expression in precursor adipocytes by 81.75% (Figure 2B). Based on these results, 5 μg/mL of polybrene and adenovirus MOI of 100 was selected as the optimal condition to induce the vectors, and the adenovirus DGAT2–shRNA–320 was selected for subsequent tests.

### 3.2. DGAT2 Affects TAG Accumulation, ADP Content, and LD Formation in Bovine Preadipocyte

As shown in Figure 3, adipocytes overexpressing *DGAT2* had significantly higher lipid, TAG, and ADP contents than those in the control group (*p* < 0.05). In contrast, *DGAT2* knockdown significantly decreased the lipid, TAG, and ADP contents of the adipocytes (*p* < 0.05).

### 3.3. DGAT2 Affects the Differentiation of Bovine Preadipocytes

To determine the effect of *DGAT2* on bovine preadipocyte differentiation, the mRNA and protein expression levels of genes related to adipose differentiation, lipid metabolism, and fatty acid synthesis were examined. As shown in Figure 4A–C, compared with those in the control group, Ad–DGAT2–infected cells had significantly higher expression levels of *PPARγ*, *C/EBPα*, *C/EBPβ*, sterol regulatory element-binding protein (*SREB*) *F1*, and fatty acid binding protein 4 (*FABP4*) (*p* < 0.05). Similarly, *DGAT2* overexpression significantly upregulated the expression of TAG synthesis-related genes [*DGAT1, DGAT2, LPIN1*, and glycerol-3-phosphate acyltransferase 4 (*GPAT4*)] and fatty acid synthesis-related genes [acetyl-CoA carboxylase alpha (*ACACA*), fatty acid synthase *(FASN*), and stearoyl-CoA desaturase: *SCD*)] (*p* < 0.05), but decreased mannosyl (alpha-1,3-)-glycoproteinbeta-1,2-N-acetylglucosaminyltransferase 1 (*MGAT1*) expression (*p* < 0.05). Consistent with RNA-seq data, *DGAT2* overexpression increased the protein levels of *DGAT1*, *DGAT2*, *PPARγ*, *C/EBPα*, *C/EBPβ*, and *LPIN1* (Figure 4D and Appendix A). In contrast, *DGAT2* knockdown significantly downregulated the expression of adipocyte differentiation-related genes (*PPARγ*, *C/EBPα*, and *C/EBPβ*) and TAG synthesis-related genes (*DGAT2*, 1-acylglycerol-3-phosphate O-acyltransferase 4, *MGAT1*, *LPIN1,* and *ACACA*) (*p* < 0.05). However, the mRNA expressions of *SREBF1*, *FABP4*, *DGAT1*, *GPAT4*, and *FASN* were significantly upregulated (*p* < 0.05) (Figure 5A–C). Moreover, *DGAT2* knockdown decreased the protein levels of *DGAT2*, *PPARγ*, *C/EBPα*, *C/EBPβ,* and *LPIN1* (Figure 5D and Appendix A).

### 3.4. Differential Analysis of Ad–DGAT2/sh–DGAT2 Infected Bovine Preadipocytes

The total RNA integrity test showed that the quality of extracted RNA met the requirements for library construction and sequencing (Figure 6A). The quality evaluation results of sequencing data showed that high-quality clean reads were >88.00% in all groups (Appendix A). The libraries were aligned against the Bos taurus (https://ftp.ensembl.org/pub/release-86/gtf/bos_taurus/Bos_taurus.UMD3.1.86.gtf.gz (accessed on 25 December 2021)) genome (Appendix A). The average degree of gene coverage showed that the data had no obvious bias (Figure 6B), and the log10 (FPKM + 1) showed normal distribution (Figure 6C).

Differential expression analysis identified 208 DEGs in the Ad–DGAT2 vs. Ad–NC groups, among which 106 were upregulated, and 102 were downregulated DEGs. Additionally, a total of 378 DEGs were identified in the sh–DGAT2 vs. sh–NC groups, among which 230 were upregulated, and 148 were downregulated DEGs. Upregulated genes are shown in red, whereas downregulated genes are shown in blue (Figure 7).

### 3.5. GO Function Analysis

The biological functions of DEGs were elucidated by GO enrichment analysis, and the 20 most significantly enriched GO terms were ranked according to the significance level. Most of the DEGs in the Ad–DGAT2 vs. Ad–NC groups were enriched in biological processes, including lipid metabolic, fatty acid biosynthetic, lipid biosynthetic, and cellular lipid metabolic processes (Figure 8A). Moreover, most of the DEGs in the sh–DGAT2 vs. sh–NC groups were enriched in the extracellular region, plasma membrane, and vesicle in the “Cellular Component” category and in cell adhesion, biological adhesion, cell surface receptor signalling pathway in the “Biological Process” category (Figure 8B).

### 3.6. KEGG Pathway Enrichment Analysis

KEGG pathway enrichment analysis showed that most DEGs in the Ad–DGAT2 vs. Ad-NC groups were enriched in the PPAR signalling pathway, cholesterol metabolism, cAMP signalling pathway, fatty acid biosynthesis, AMP-activated protein kinase (AMPK) signalling pathway, and unsaturated fatty acid biosynthesis (Figure 9A). Particularly, seven DEGs were enriched in the PPAR signalling pathway (Figure 9B), including four upregulated genes [(acyl-CoA desaturase, acyl-CoA synthetase long-chain family member *(ACSL) 3*, fatty acid desaturase 2 (*FADS2*), and 3-hydroxy-3-methylglutaryl-CoA synthase 1 (*HMGCS1*)] and three downregulated genes [(*ACSL6*; angiopoietin-like 4 (*ANGPTL4*) and apolipoprotein A 1(*APOA1*)]. Four DEGs were enriched in the cholesterol metabolism signalling pathway (Figure 9C), including one upregulated gene (ATP-binding cassette family A1) and three downregulated genes (*ANGPTL4*, *APOA1*, and *APOA4*). Five DEGs were enriched in the AMPK signalling pathway (Figure 9D), including four upregulated genes [(acyl-CoA desaturase, *SREBF1*, *FASN*, and solute carrier family 2 (facilitated glucose transporter)member 4 (*SLC2A4*)] and one downregulated gene (factor binding protein 1).

DEGs in the sh–DGAT2 vs. sh–NC groups were mostly enriched in pathways related to human diseases. However, some DEGs were enriched in pathways related to adipocyte differentiation, including Hippo, transforming growth factor (TGF)–β, and cAMP signalling pathways (Figure 10A). Specifically, two upregulated DEGs (naked cuticle homolog 1 and growth differentiation factor 6 (*GDF6*)] and two downregulated DEGs (crumbs cell polarity complex component 2 and integrin beta 2) were enriched in the Hippo signalling pathway (Figure 10B). Additionally, three upregulated DEGs (inhibin βE, *GDF6*, and Noggin) were enriched in the TGF–β signalling pathway, whereas three upregulated DEGs [(gastric inhibitory polypeptide receptor, endothelin 1 (*EDN1*), and troponin I3)] (Figure 10C) and one downregulated DEG (*EDN2*) were enriched in the cAMP signalling pathway (Figure 10D).

## 4. Discussion

The ADVs have advantages, such as a wide host range, high infection efficiency, and low genome rearrangement probability, and they have been widely used in gene expression studies. TAGs are the main form of energy storage in eukaryotic cells, and most TAGs are mainly stored in fat cells. As a key rate-limiting enzyme in TAG synthesis, *DGAT2* plays an important role in lipid accumulation [12], and its expression directly determines the differentiation ability of fat precursor cells. *DGAT2* has been shown to regulate the accumulation of TAGs in the tissues of DGAT1–deficient mice [8]. Additionally, *DGAT2* was highly expressed in various lipid-metabolising tissues [13]. Moreover, *DGAT2* can compensate for LD formation in DGAT1–deficient intestinal stem cells [14]. In the present study, *DGAT2* overexpression increased the formation and accumulation of LD in Yanbian cattle preadipocytes, whereas *DGAT2* knockdown inhibited LD formation in the cells, which was consistent with previous findings [15]. Several studies have shown that low expression of *DGAT2* can cause a decrease in the TAG content of adipocytes [9]. *DGAT2* overexpression increased the expression of lipid-forming genes and the accumulation of TAGs in skeletal muscle cells (BSCs) [16]. In the present study, *DGAT2* expression was positively correlated with preadipocyte TAG and ADP contents.

*GPAT4* and *LPIN1* have been identified as important regulatory factors of lipid deposition through the glycerophospholipid metabolic pathway [17]. Interference with *LPIN1* inhibited adipose tissue development and significantly reduced adipose tissue quality, whereas overexpression of *LPIN1* in skeletal muscle or adipose tissue promoted obesity in mice [18]. In the present study, both *DGAT2* overexpression and knockdown increased *GPAT4* expression, whereas *DGAT2* knockdown suppressed *LPIN1* expression. These results suggest that there may be a positive regulatory relationship between *DGAT2* and *LPIN1*. *SCD1* and *DGAT2* are located near the endoplasmic reticulum, and *SCD* is thought to mediate the de novo conversion of monounsaturated fatty acids by *DGAT2* [19]. *FABP4* is an adipogenic factor that is upregulated during the differentiation of preadipocytes [20]. In the present study, *DGAT2* overexpression significantly upregulated the mRNA expression of *PPARγ*, *C/EBPα*, *C/EBPβ*, *FABP4*, and *SREBF1*, and TAG synthesis-related genes (*GPAT4* and *LPIN1*). However, most of the genes were downregulated in sh–DAGT2–treated cells. The siRNA–mediated downregulation of goose *DGAT2* has been reported to reduce the expression of adipogenesis-related genes [21].

Several studies have shown that there is a feedback mechanism between *DGAT2* activity and the *SREBP1*–mediated pathway, which establishes a new link between *DGAT2* and adipogenesis [22]. Overexpression of DGAT2 in 3T3-L1 preadipocytes showed a higher ratio of monounsaturated fatty acids (C16:1 and C18:1), which may be due to de novo synthesis of fatty acids rather than the uptake of specific fatty acids from the medium [23]. *DGAT2*–overexpressing cells have been shown to have significantly higher expression of fatty acid synthesis genes (*ACACA*, *FASN*, and *SCD*) at both the mRNA and protein levels compared to that in the control group [16]. Similarly, precursor adipocytes overexpressing *DGAT2* had significantly high mRNA expression levels of *ACACA*, *FASN,* and *SCD* in the present study; however, the expression of *ACACA* and other genes was significantly inhibited in sh–DGAT2–transfected cells.

Transcriptome sequencing technology is an important tool for exploring gene function [24] and plays an important role in elucidating the molecular mechanisms of cells, tissues, growth, development, and disease [25]. In the present study, differential expression analysis identified 208 DEGs in the Ad–DGAT2 vs. Ad–NC groups, and GO functional annotation showed that the 20 most enriched terms included lipid metabolism, fatty acid biosynthesis, lipid biosynthesis, and cellular lipid metabolism. Moreover, KEGG pathway analysis showed that the DEGs were mainly involved in the PPAR signalling pathway, cholesterol metabolism pathway, and AMPK signalling pathway. The PPAR signalling pathway plays an important role in regulating adipogenesis, fatty acid transport, and adipocyte differentiation. Acyl-CoA desaturase, *ACSL3*, *FADS2*, and *HMGCS1* were enriched in the PPAR signalling pathway. *ACSLs* are involved in the initial step of almost all metabolic pathways in mammals, including complex lipid synthesis, protein modification, and fatty acid beta-oxidation. *ACSL3*, a member of the *ACSLs* family, contains high levels of polyunsaturated fatty acids called phospholipids, which are widely expressed in the endoplasmic reticulum and LD of most tissues. *ACSL3* can regulate the activity of key transcription factors for adipocyte differentiation, help maintain lipid balance, and participate in the absorption of fatty acids. *ACSL3* can promote the synthesis of TAG and LD in cells, promote the expression of adipogenic differentiation marker genes, and positively regulate adipocyte differentiation in Chinese grassland Red Bulls. Moreover, a g.111870595G > A(E6-G > A) mutation in the coding region of *ACSL3* is correlated with fatty acid content and some meat quality traits [26].

The AMPK signalling pathway plays a key role in the metabolism pathway by regulating energy homeostasis and cellular lipid metabolism [27]. *DGAT2* overexpression can upregulate lipid synthesis-related genes (*SREBF1*, *FASN,* and *SLC2A4*) downstream of AMPK to regulate adipogenesis. *FASN* is crucial for breast functional development and milk fat production and plays an important role in fatty acid biosynthesis [28]. *SREBP1* regulates milk fat synthesis and secretion [29]. *SREBP1* overexpression can increase the expression of *FASN* and *SREBF1* in primary goat mammary epithelial cells [30]. *SREBP1* is a regulator of *FASN* gene expression, and *SREBF* chaperone (*SCAP*) regulates the activity of *SREBP1*. *SCAP* is essential to promote nuclear translocation of SREBP1 and activate transcription of the *FASN* gene, thus promoting LD formation in bovine mammary epithelial cells.

KEGG analysis of 378 DEGs in the sh–DGAT2 vs. sh–NC groups showed that most of the DEGs were involved in human diseases. However, some DEGs were enriched in pathways related to adipocyte differentiation, such as Hippo, TGF-β, and cAMP signalling pathways. The Hippo signalling pathway inhibits adipogenesis by inhibiting the transcriptional activity of *PPARγ*, a key factor in adipogenesis. Therefore, we hypothesised that the inhibition of adipocyte differentiation in sh-DGAT2-treated cells could be attributed to the activation of the Hippo signalling pathway, which inhibited the expression of *PPARγ* and suppressed the differentiation of precursor adipocytes. TGF-β helps limit the differentiation potential of cells, inhibiting the differentiation of osteoblasts, skeletal muscle myoblasts, and adipocytes and stimulating the proliferation of these cells before they reach full maturity. A TGF-β/Smads signalling pathway is one of the key pathways in which TGF-β plays a major role in adipogenesis by activating MAPK and P13K/Akt signalling pathways through non-SMAD signalling pathways [31].

Identifying candidate genes involved in preadipocyte differentiation and proliferation would promote the breeding of Yanbian cattle. Overall, the results of this study validated the regulatory role of *DGAT2* in adipocyte differentiation and lipid metabolism and provided a theoretical basis for further improvement of bovine genome annotation and molecular breeding.

## 5. Conclusions

In summary, the present study showed that *DGAT2* overexpression promoted preadipocyte differentiation, LD production, and TAG accumulation by increasing the expression of fat metabolism-related genes and proteins. Differential expression analysis identified 208 and 378 DEGs in the Ad–DGAT2 vs. Ad–NC and sh–DGAT2 vs. sh–NC groups, most of which were enriched in AMPK, TGF-β, and PPAR signalling pathways. These pathways regulate adipocyte proliferation, differentiation, as well as lipid production by regulating the transcription of some genes in adipocytes.

## Figures and Tables

**Figure 1 animals-13-01195-f001:**
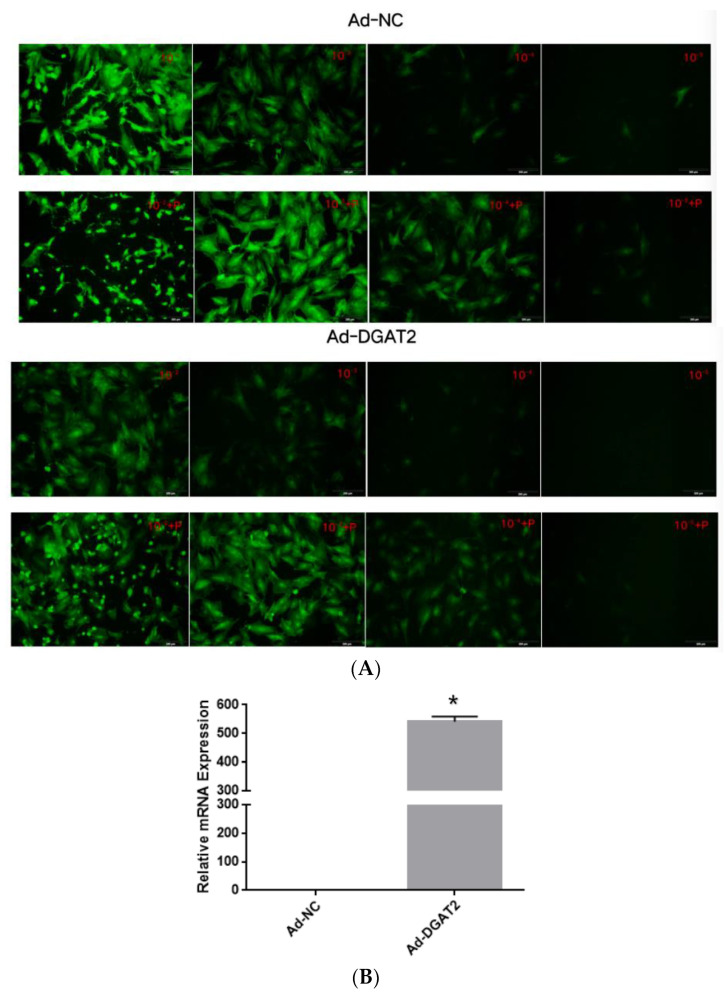
Green fluorescence expression of bovine preadipocytes infected with high-titre adenovirus Ad–NC and Ad–DGAT2 (**A**) for 48 h. (**B**) *DGAT2* mRNA expression. Scale bars = 200 µm. The results were presented as the mean ± SEM (*n* = 3) * *p* < 0.05.

**Figure 2 animals-13-01195-f002:**
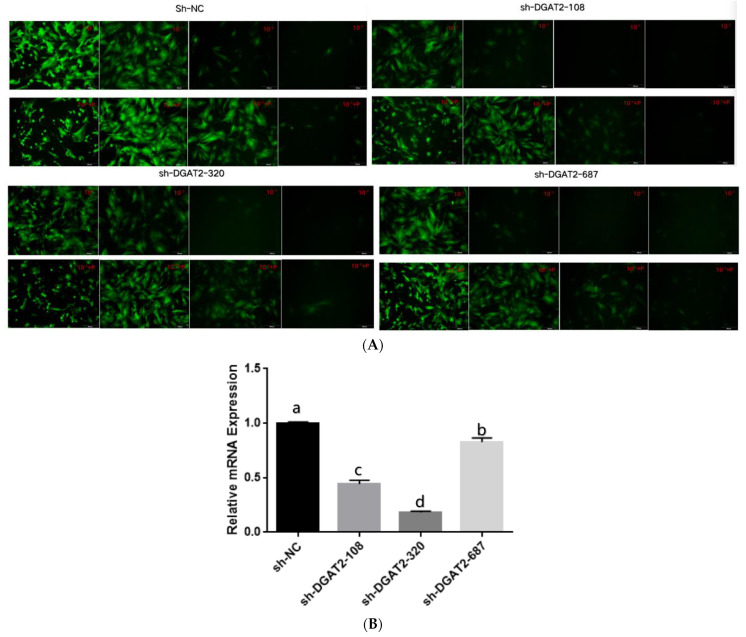
Green fluorescence expression of bovine preadipocytes infected with high-titre adenovirus sh–NC, sh–DGAT2–108, sh–DGAT2–320 and sh–DGAT2–687 (**A**) for 48 h, and (**B**) DGAT2 mRNA expression. Scale bars = 200 µm. The results were presented as the mean ± SEM (*n* = 3). The different letters (a–d) represent significant differences (*p* < 0.05) in gene expression.

**Figure 3 animals-13-01195-f003:**
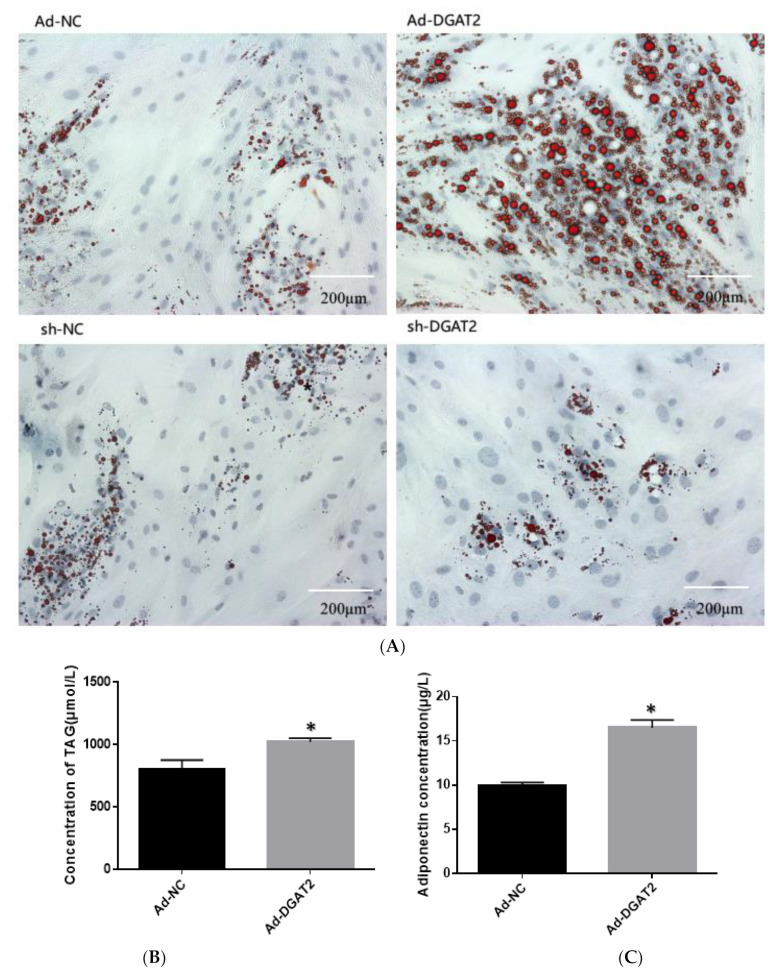
*DGAT2* affects the lipid accumulation and adiponectin (ADP) content of adipocytes. (**A**) Oil red O staining (scale bar = 200 um). (**B**,**D**) Triacylglycerol (TAG) concentration. (**C**,**E**) ADP concentration in the medium. The results were presented as the mean ± SEM (*n* = 3). * *p* < 0.05.

**Figure 4 animals-13-01195-f004:**
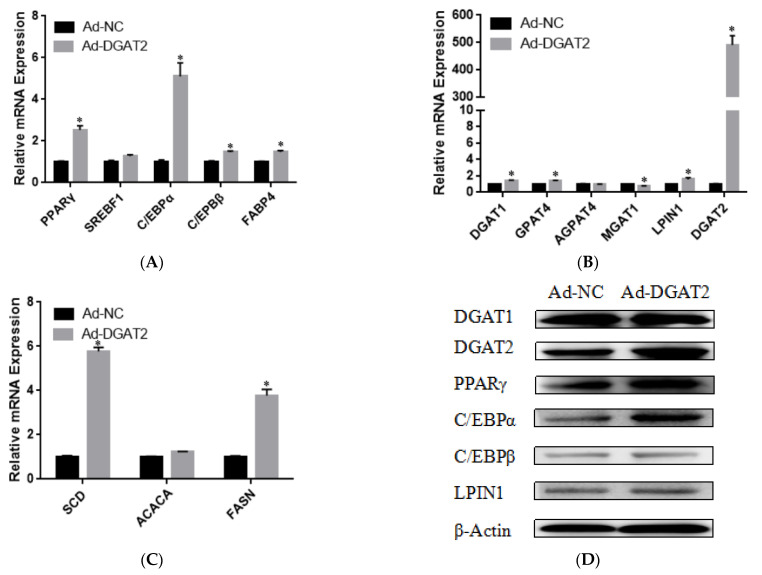
Overexpression of *DGAT2* affected the expression of (**A**) adipose differentiation, (**B**) lipid metabolism, (**C**) and fatty acid synthesis-related genes at both the mRNA and (**D**) protein levels. Values are presented as means ± SEM (*n* = 3). * *p* < 0.05.

**Figure 5 animals-13-01195-f005:**
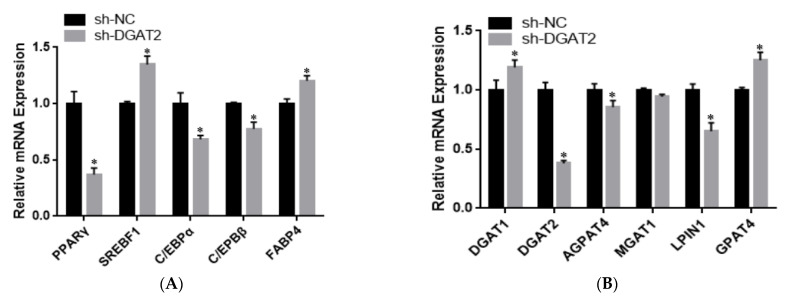
*DGAT2* knockdown affected the expression of adipose differentiation (**A**), lipid metabolism (**B**), and fatty acid synthesis (**C**) related genes at both the mRNA and protein levels (**D**). Values are presented as means ± SEM (*n* = 3) * *p* < 0.05.

**Figure 6 animals-13-01195-f006:**
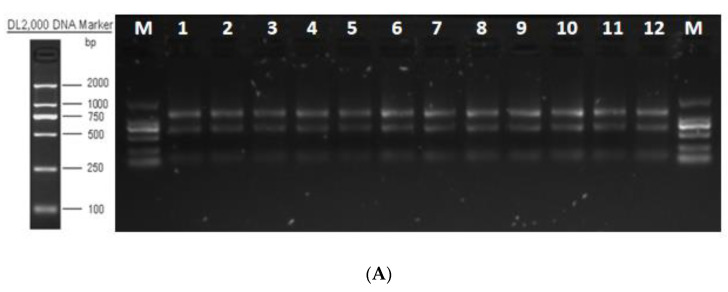
Total RNA quality detection and RNA-seq data quality assessment. (**A**) Total RNA integrity detection. (**B**) Gene coverage uniformity. (**C**) Density distribution of FPKM.

**Figure 7 animals-13-01195-f007:**
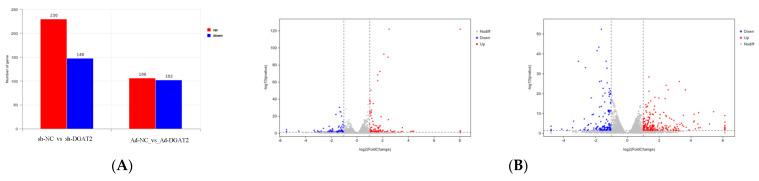
RNA–seq–based differentially expressed genes (DEGs) between the groups. (**A**) Statistics of DEGs in the groups. (**B**) Volcano map of DEGs. Significantly upregulated DEGs are indicated with red dots, whereas downregulated DEGs are represented with blue dots. Grey dots are non significant DEGs.

**Figure 8 animals-13-01195-f008:**
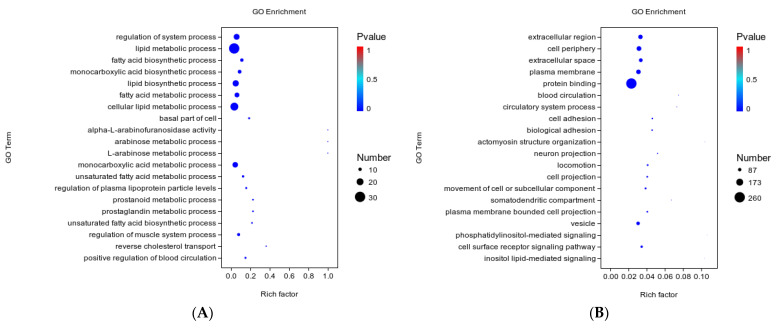
Gene ontology (GO) term analysis of the differentially expressed genes (DEGs). GO functional annotation of the DEGs in biological processes, cellular components, and molecular functions. The 20 significantly enriched GO terms. (**A**) GO functional annotation of DEGs in the Ad–DGAT2 vs. Ad–NC groups (**B**) and the sh–DGAT2 vs. sh–NC groups.

**Figure 9 animals-13-01195-f009:**
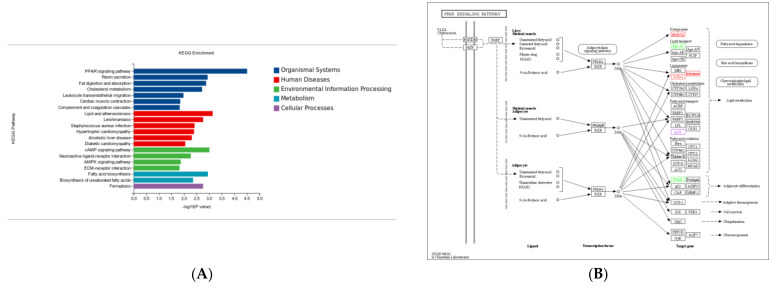
Kyoto Encyclopaedia of Genes and Genomes (KEGG) pathway analysis of differentially expressed genes (DEGs) in the Ad–DGAT2 vs. Ad–NC groups. (**A**) KEGG enrichment histogram. (**B**–**D**) DEGs were enriched in PPAR, cholesterol metabolism, and AMPK signalling pathways.

**Figure 10 animals-13-01195-f010:**
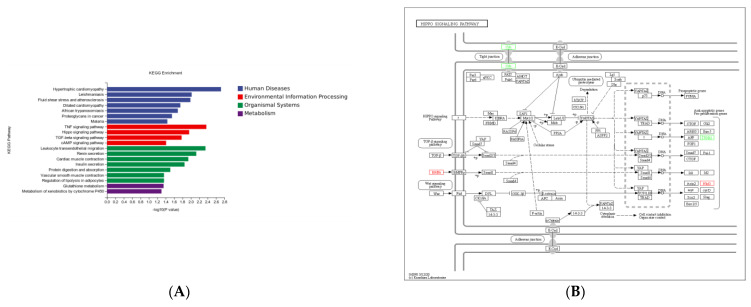
Kyoto Encyclopaedia of Genes and Genomes (KEGG) pathway analysis of differentially expressed genes (DEGs) in the sh–DGAT2 vs. sh–NC groups. (**A**) KEGG enrichment histogram. (**B**–**D**) DEGs were enriched in Hippo, TGF–β, and cAMP signalling pathways.

**Table 1 animals-13-01195-t001:** shRNA and sh–NC sequences for the *DGAT2* mRNA.

Scheme 5	Sense Strand (5′-3′)	Anti-Sense Strand (3′-5′)	Target Sequence
DGAT2-sh-108	AATTCGGTAGAGAAGCAGCTCCAAGTTTCAAGAGAACTTGGAGCTGCTTCTCTACCTTTTTTG	GATCCAAAAAAGGTAGAGAAGCAGCTCCAAGTTCTCTTGAAACTTGGAGCTGCTTCTCTACCG	GGTAGAGAAGCAGCTCCAAGT
DGAT2-sh-320	AATTCGCTACTTTCGAGACTACTTTCTTCAAGAGAGAAAGTAGTCTCGAAAGTAGCTTTTTTG	GATCCAAAAAAGCTACTTTCGAGACTACTTTCTCTCTTGAAGAAAGTAGTCTCGAAAGTAGCG	GCTACTTTCGAGACTACTTTC
DGAT2-sh-687	AATTCGCGCAATCGCAAGGGCTTTGTTTCAAGAGAACAAAGCCCTTGCGATTGCGCTTTTTTG	GATCCAAAAAAGCGCAATCGCAAGGGCTTTGTTCTCTTGAAACAAAGCCCTTGCGATTGCGCG	GCGCAATCGCAAGGGCTTTGT
sh-NC	AATTCGTTCTCCGAACGTGT CACGTTTCAAGAGAACGTGAC ACGTTCGGAGAACTTTTTTG	GATCCAAAAAAGTTCTCCGA ACGTGTCACGTTCTCTTGAA ACGTGACACGTTCGGAGAACG	TTCTCCGAACGTGTCACGT

**Table 2 animals-13-01195-t002:** Sequence information of PCR primers.

Gene	Sense Strand (5′-3′)	Length (bp)	Gene ID
*GAPDH*	F:ACTCTGGCAAAGTGGATGTTGTC R:GCATCACCCCACTTGATGTTG	143	NM_001034034
*DGAT1*	F:CTACACCATCCTCTTCCTCAAG R:AGTAGTAGAGATCGCGGTAGGTC	176	NM_174693.2
*DGAT2*	F:GACCCTCATAGCCTCCTACTCC R:GACCCATTGTAGCACCGAGATGAC	145	NM_205793.2
*AGPAT4*	F:TGTTCTCGTCTTCTTTGTGGCTTCC R:TCGCTATGTTTCTGCTTGCTGTCC	111	NM_001015537.1
*MGAT1*	F:AGCCGTGGTGGTAGAGGATGATC R:TGCTCCTTGCCATTGTCGTTCC	132	NM_001015653
*LPIN1*	F:AGTCCTCGCCACACAAGATG R:AGATGCCCTGACCAGTGTTG	137	NM_001206156
*GPAT4*	F:ATGCGGTCCAGTTTGCCAATAGG R:GCTTCTGCTGCTCCTCCTTGAAC	129	NM_001083669.1
*PPARγ*	F-ATCTGCTGCAAGCCTTGGA R-TGGAGCAGCTTGGCAAAGA	138	NM_181024
*C/EBPα*	F-CCAGAAGAAGGTGGAGCAACTG R-TCGGGCAGCGTCTTGAAC	69	NM_176788
*C/EBPβ*	F-CAACCTGGAGACGCAGCACAAG R-CGGAGGAGGCGAGCAGAGG	143	NM_176788
*SREBF1*	F-CTGCTGACCGACATAGAAGACAT R-GTAGGGCGGGTCAAACAGG	81	NM_001113302
*FABP4*	F-AAACTTAGATGAAGGTGCTCTGG R-CATAAACTCTGGTGGCAGTGA	134	NM_174314.2
*FASN*	F-CGCTTGCTGCTGGAGGTCAC R-GGTCTCAGGGTCTCGGCTCAG	141	NM_001012669
*SCD*	F-TGCCCACCACAAGTTTTCAG R-GCCAACCCACGTGAGAGAAG	80	NM_173959
*ACACA*	F-GCCAAACCTCTGGAGCTGAA R-CGAGCTTCACCAGGTTGCTA	97	NM_174224

## Data Availability

The datasets generated and/or analyzed during the conduct of the study are included in this published article. The raw RAN-Seq data has been uploaded to the SRA database, and the bioproject accession is PRJNA949284.

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
