# Peer review of "Overexpression of DGAT2 Regulates the Differentiation of Bovine Preadipocytes"

_animals, 2023, doi:10.3390/ani13071195_

Round 1

Reviewer 1 Report

 In my opinion the manuscript is interesting and worth publishing, but its current form prevents me from recommending it for publication. I must point out, that I am able to evaluate only genetic part of the experiment, and another reviewer is necessary for remaining part of analyses.

1.            The language of manuscript needs to be revised by a native speaker. There are several grammatical and stylistic errors that need to be improved. Even though I am not a native speaker, the text in some parts is difficult to read for me and sounds unprofessional.

2.            In the introduction section there is hard to tell if authors are saying about DGAT gene or protein. Please use italics for gene name.

3.            Description of RNA-Seq analysis needs to be improved. It is too brief and lack substantial information. Please add information on library preparation it (HiSeq 2000 is a sequencing system not a library prep kit), reads length and run setup. Also please supplement the information on data analysis process. It lacs information on reads filtering, mapping and counting. Finally, how many reads were generated per sample? Please also include mapping statistics, including information on uniquely mapped reads, reads mapped to genes, and number of expressed genes. Please provide also version of the genome used for mapping (provided link shows only basic ncbi genome site).

4.            In the methods section concerning statistics, please clarify to which data this analysis applies. Please clarify also what does it means ‘three independent experiments’ (technical replicates?). Did you tested data distribution (normality) and equality of variances for AVOVA?

5.            Description of figures 6, 7 and 8 is vague. Please provide more details. Describe what values are presented on graphs. The axes descriptions are hardly readable. In Figure 7, blue color on volcano plot shows genes downregulated, not of low expression. Similarly, for red color. This is correct in Figure caption, but not in the text above.

I recommend major revision of the manuscript.

Author Response

Point 1: The language of manuscript needs to be revised by a native speaker. There are several grammatical and stylistic errors that need to be improved. Even though I am not a native speaker, the text in some parts is difficult to read for me and sounds unprofessional. 

Response 1: Thank you for the keen review of our manuscript. Our manuscript has been polished again by Editage (www.editage.cn). The following is the editing certificate.

Point 2: In the introduction section there is hard to tell if authors are saying about DGAT gene or protein. Please use italics for gene name. 

Response 2: Thank you for the thorough review of our manuscript. We have revised the preface and reviewed the manuscript to ensure that all the gene name are in italics.

Point 3: Description of RNA-Seq analysis needs to be improved. It is too brief and lack substantial information. Please add information on library preparation it (HiSeq 2000 is a sequencing system not a library prep kit), reads length and run setup. Also please supplement the information on data analysis process. It lacs information on reads filtering, mapping and counting. Finally, how many reads were generated per sample? Please also include mapping statistics, including information on uniquely mapped reads, reads mapped to genes, and number of expressed genes. Please provide also version of the genome used for mapping (provided link shows only basic ncbi genome site). 

Response 3: Thanks for your comment. We have further refined the description of RNA-Seq analysis.

Point 4: In the methods section concerning statistics, please clarify to which data this analysis applies. Please clarify also what does it means ‘three independent experiments’ (technical replicates?). Did you tested data distribution (normality) and equality of variances for AVOVA?

Response 4: Thanks for your comment. We have refined the description of the methods section concerning statistics.

Point 5: Description of figures 6, 7 and 8 is vague. Please provide more details. Describe what values are presented on graphs. The axes descriptions are hardly readable. In Figure 7, blue color on volcano plot shows genes downregulated, not of low expression. Similarly, for red color. This is correct in Figure caption, but not in the text above.

Response 5: Thanks for your comment. We have supplemented the details of figures 6, 7 and 8 .Thank you very much.

Reviewer 2 Report

The manuscript from Guo et al. entitled "Overexpression of DGAT2 regulates the differentiation of bovine preadipocytes" is a very interesting molecular and transcriptomic investigation of the mechanism of DGAT2 in lipid metabolism and preadipocyte differentiation from Yanbian cattle. The manuscript is well organized, the cited literature is relevant to the questions discussed and the figures are well made. However, before recommending it for publication, I am sending a few comments below, which in my opinion should be taken into account by the authors:

1. Line 82 – It is not clear what means “1% PBS”. Please be more specific.

2. Line 88 – What is anti-growth medium? Please provide more information.

3. Line 90 – I suggest the change of “differentiation medium containing 100 µM of palmitoleic acid (PA), 5% FBS, and DMEM” with “differentiation medium (DMEM supplemented with 5% FBS and 100 µM palmitoleic acid)”

4. Line 123-125 – “Before infection, the cells were transferred into 90 μL of fresh culture medium containing 10 μL of virus at different concentration gradients, and each group had three multiple wells.” This means that the adherent cells were detached by trypsinization, washed by centrifugation and then reseeded in 96-well plates containing 90 μL of fresh culture medium and 10 μL of virus at different concentration gradients? Or, the culture medium of each wells were changed with 90 μL of fresh culture medium and 10 μL of virus at different concentration gradients? Please be clearer.

5. Line 140-142 – The reaction conditions of PCR are not clear. Please revise!

6. Line 226 – At figure 2, the GFP expression of bovine preadipocytes are not infected with Ad-NC and Ad-DGAT2 (A). Please correct with Sh-NC, Sh-DGAT2-108, Sh-DGAT2-320 and Sh-DGAT2-687 (A).

7. Line 244 – At figure 3, please add “Results are presented as the mean ± standard error of (SEM) of three individual cultures. * p < 0.05.”

Author Response

Point 1: Line 82 – It is not clear what means “1% PBS”. Please be more specific.

Response 1: Thanks for your comment. We have made a complete expression of “1% PBS”(Line 86-87).

Point 2: Line 88 – What is anti-growth medium? Please provide more information.

Response 2: Thank you very much for your comments. We've switched to another expression(Line 92).

Point 3: Line 90 – I suggest the change of “differentiation medium containing 100 µM of palmitoleic acid (PA), 5% FBS, and DMEM” with “differentiation medium (DMEM supplemented with 5% FBS and 100 µM palmitoleic acid)”

Response 3: Thank you for the keen review of our manuscript. We have replaced it according to your suggestion(Line 94).

Point 4: Line 123-125 – “Before infection, the cells were transferred into 90 μL of fresh culture medium containing 10 μL of virus at different concentration gradients, and each group had three multiple wells.” This means that the adherent cells were detached by trypsinization, washed by centrifugation and then reseeded in 96-well plates containing 90 μL of fresh culture medium and 10 μL of virus at different concentration gradients? Or, the culture medium of each wells were changed with 90 μL of fresh culture medium and 10 μL of virus at different concentration gradients? Please be clearer.

Response 4: Thank you for the thorough review of our manuscript. We have explained it clearly (Line 132-134).

Point 5: Line 140-142 – The reaction conditions of PCR are not clear. Please revise!

Response 5: Thanks for your comment. We have revised the reaction conditions of PCR (Line 149-151).

Point 6: Line 226 – At figure 2, the GFP expression of bovine preadipocytes are not infected with Ad-NC and Ad-DGAT2 (A). Please correct with Sh-NC, Sh-DGAT2-108, Sh-DGAT2-320 and Sh-DGAT2-687 (A).

Response 6: Thank you for the thorough review of our manuscript. Ad-NC and Ad-DGAT2 have been corrected by sh-NC, sh-DGAT2-108, sh-DGAT2-320, and sh-DGAT2-687 (Line 259-261).

Point 7: Line 244 – At figure 3, please add “Results are presented as the mean ± standard error of (SEM) of three individual cultures. * p < 0.05.”

Response 7: Thanks for your comment. We have added the relevant information (Line 280-281).

Round 2

Reviewer 1 Report

The manuscript was significantly improved, but it steel needs correction regarding description of RNA-Seq experiment. Despite several information was supplemented, methodology still lacs information about read length (2x150bp maybe?), which software was used for mapping, what was the genome version, which annotation file was used and others. Please see some other publications performing RNA-Seq and prepare similar description. Please also put the raw or processed RAN-Seq data to public archives, such as SRA or GEO databases.

Author Response

Thank you again for the keen review of our manuscript. We have supplemented the content of the description of RNA-Seq experiment, and the   raw RAN-Seq data is being uploaded to the SRA database. Thank you very much!
